# OpenReview forum: "Toward Unified Robot Learning: Bridging Representation, Vision-Language-Action, and World Models"
_TMLR — Under review for TMLR_

### Review · Reviewer_H9Fq · 2026-05-10

**Summary Of Contributions:**

This paper presents a comprehensive survey of the current state of robot learning, proposing a unified framework that organizes the field into three primary pillars:

1. Representation Learning: How robots perceive and encode high-dimensional sensory data.

2. Vision-Language-Action (VLA) Models: How robots map perceptions and instructions directly to executable actions.

3. World Models: How robots reason about the dynamics of their environment and predict future outcomes.

Key Strengths:

1. Structured Taxonomy: It moves beyond a simple list of papers to provide a conceptual "three-axis" framework (Understanding, Acting, Reasoning).

2. Interdisciplinary Bridge: Effectively connects computer vision, NLP (LLMs/VLMs), and control theory.

3. Future-Facing: Identifies critical bottlenecks such as cross-embodiment transfer and long-horizon planning, offering a roadmap for "Unified Robot Learning."

Key Weaknesses:

1. Depth vs. Breadth: As a "Long Submission," the paper covers vast ground, which may occasionally lead to a high-level treatment of specific algorithmic nuances in favor of broad categorization.

2. Hardware Realism: While focusing on "physical grounding," the transition from simulation-heavy world models to noisy, real-world hardware constraints could be explored with more empirical depth.

**Additional Comments:**

The paper is well-written and serves as a timely intervention. I lean to accept this survey paper.

**Audience:**

Yes

**Audience Explanation:**

Robot learning is a rapidly expanding subfield of machine learning. Given TMLR's interest in fundamental ML research, a survey that attempts to unify disparate paradigms (Self-Supervised Learning, Generative Modeling, and Reinforcement Learning) into a single robotic framework is highly relevant. It provides a "state-of-the-union" for researchers looking to enter the field or for specialists seeking to understand how their work fits into the broader "Generalist Robot" objective.

**Broader Impact Concerns:**

As a survey paper, this work does not introduce new sensitive datasets or specific dual-use technologies. It focuses on the organizational and theoretical aspects of robot learning. The authors briefly touch upon "reliable operation" and "robustness," which are inherently safety-positive goals for real-world robotics

**Claims And Evidence:**

Yes

**Claims Explanation:**

The authors support their taxonomy by citing a wide array of seminal and recent works. The claim that these fields are currently "fragmented" is well-supported by the analysis of how representation learning often ignores action-trace dynamics, and how VLAs often lack the explicit temporal reasoning found in world models. The paper's evidence is primarily bibliographic and conceptual, which is appropriate for a survey of this scale.

**Requested Changes:**

1. Uncertainty Quantification: Strengthen the discussion on how "Probabilistic Robot Learning" can be practically implemented within VLA architectures, as these are often treated deterministically.

2. Benchmark Comparison: Include a summary table or visualization comparing the datasets and benchmarks used across these three axes (e.g., BridgeData, Open X-Embodiment) to help readers understand where the data "gravity" currently lies.

3. Clarification on "World Models": Distinguish more clearly between "Latent World Models" (like Dreamer) and "Video Generation Models" (like SORA-style approaches) used as world simulators.

---

> ### Author Response · Authors · 2026-06-23
> **Author Response**
>
> We thank the reviewer for their feedback. We carefully revised the manuscript based on your suggestions. Please find our responses below (The updated text is highlighted in blue in the revised manuscript)
>
> **[Q1] Uncertainty Quantification: Strengthen the discussion on how "Probabilistic Robot Learning"**
>
> We thank the reviewer for this suggestion. We agree that the original discussion of uncertainty quantification was too abstract and did not sufficiently explain how probabilistic robot learning could be implemented within modern VLA architectures, which are often deployed as deterministic action predictors.
> To address this, we now connect probabilistic robot learning to uncertainty-aware VLA action generation in the ‘Discussion and Challenges’ section of Section 5. The revised text explains that although generative action heads can represent multiple plausible robot behaviors, many VLA systems still execute the most likely action or trajectory without explicitly reasoning about reliability under visual perturbations, instruction ambiguity, occlusions, or sensor noise. We have also added some implementation mechanisms that can be incorporated into existing VLA systems and cite recent works that use uncertainty estimates in VLA models, including approaches that use token-level uncertainty to trigger human intervention when confidence falls below a threshold.
> Finally, we revised the discussion to emphasize that these mechanisms remain challenging to deploy in closed-loop robot control. We believe this addition makes the uncertainty discussion more practically connected to VLA architectures while also highlighting the remaining deployment challenges.
>
> **[Q2] Benchmark Comparison**
>
> We thank the reviewer for this helpful suggestion. We agree that the original manuscript did not sufficiently summarize the dataset and benchmark landscape across the three axes of the survey. In the revised manuscript, we added a dataset/benchmark comparison table in Section 2 (Table 3) to help readers understand where the current “data gravity” lies.
> The new table summarizes representative datasets and benchmarks according to their modalities, robot embodiment coverage, and relevance to the three axes of the survey. The table also indicates the typical role each dataset plays, for example whether it primarily supports learning environment representations, training robot policies from demonstrations, evaluating language-conditioned robot behavior, or learning predictive models from trajectories and videos.
> We also added surrounding text clarifying that datasets shape how representation learning, VLA models, and world models are studied. Some datasets concentrate data around large-scale robot demonstrations and therefore primarily support VLA policy learning, while others emphasize simulation, multi-view observations, trajectories, or future prediction and are more naturally suited to representation learning or world-model training. This addition makes the survey’s data landscape clearer and helps readers see which axes are currently well supported by large-scale data and which remain comparatively underdeveloped.
>
> **[Q3] Clarification on World Models**
>
> We thank the reviewer for this suggestion. We agree that the original manuscript used the term “world model” too broadly and did not clearly distinguish control-oriented latent dynamics models from video-generation models used as neural simulators.
> To address this, we added a clarification in Section 6.1 to distinguish between two related but different uses of world models in robot learning. We now describe control-oriented latent world models, such as Dreamer-style recurrent state-space models, as models that learn compact latent dynamics to support imagination, reward and value prediction, and policy optimization. We contrast this with video-generation-based world models, which use diffusion, autoregressive, or flow-based architectures to synthesize visually grounded future observations or trajectories conditioned on language, actions, or initial scene context. We also clarify that the distinction is not simply latent-space versus pixel-space prediction, since many video-generation models also operate in learned latent spaces. Instead, the key distinction is how prediction is coupled to policy learning: control-oriented latent world models are optimized to preserve task-relevant dynamics for control, while video-generation models generate visually grounded futures that must be connected to action through inverse dynamics, action decoders, rollout evaluation, or policy conditioning. We believe this addition makes the scope of “world models” clearer in the revised manuscript.

---

### Review · Reviewer_bWSw · 2026-05-26

**Summary Of Contributions:**

**Summary**

This paper presents a comprehensive survey of robot learning, framing the field around three core pillars: environment representation learning, VLA models, and generative world models. The authors establish a structured taxonomy of existing methodologies and contrast their survey against prior work to emphasize its holistic scope. They outline five fundamental open challenges: uncertainty quantification, out-of-distribution generalization, cross-embodiment transfer, long-context understanding, and long-horizon prediction. Finally, the authors analyze the limitations of current independent paradigms, run a small qualitative diagnostic experiment on world models using the CALVIN dataset, and highlight emerging trends toward unified, physically grounded systems.

**Strengths**

- The survey cleanly categorizes robot learning literature into representation, VLA models, and world models.
- The authors provide an explicit analysis contrasting deterministic collapse and generative distributions in visual policy execution. They detail the fine-grained mechanics, constraints, and algorithmic trade-offs between diffusion-based and flow-matching action models.
- The submission integrates a practical qualitative diagnostic using the CALVIN dataset to validate arguments about model shortfalls.

**Weaknesses**

I only have a few minor comments:
- The discussion on hierarchical and reasoning-augmented VLA models fails to analyze the real-time execution delay or token bottlenecks imposed by large backbones on physical hardware control loops.
- Section 3.1 outlines aleatoric and epistemic mathematical concepts but lacks an active cross-reference showing how modern VLA frameworks implement these parameters directly into real-world control safety boundaries.
- Notations like $z^{\le t}$, $c^{\le t}$ do not seem standard. Would $z^{1:t}$, $c^{1:t}$ be better?

**Audience:**

Yes

**Audience Explanation:**

Yes. TMLR's audience includes a substantial community of researchers focused on robot learning, reinforcement learning, and multimodal foundation models.

**Claims And Evidence:**

Yes

**Claims Explanation:**

Yes, the core claim that robot learning paradigms are structurally fragmented across representation, action, and world modeling is convincingly supported. The authors supply an extensive meta-review of current literature (Table 1) and map out how isolated components fail to simultaneously address major real-world challenges (Table 2). The arguments regarding the physical shortcomings of vision-only world models are grounded in a concrete diagnostic showcase on the CALVIN dataset.

**Requested Changes:**

See weaknesses

---

> ### Author Response · Authors · 2026-06-23
> **Author Response**
>
> We thank the reviewer for their suggestions. We have revised the manuscript based on your suggestions. Please find our responses below (The updated text is highlighted in blue in the revised manuscript)
>
> **[Q1] real-time execution delay or token bottlenecks imposed by large backbones**
>
> We thank the reviewer for raising this point. We agree that hierarchical and reasoning-augmented VLA models can introduce practical limitations for real-time robot control, especially when large language or multimodal backbones are used to generate intermediate goals, plans, or reasoning traces. To address this, we added a discussion of this limitation in Section 5.3. The revised text now notes that although hierarchical structures can improve long-horizon planning, they may also introduce inference latency or generation bottlenecks. We clarify the resulting trade-off between semantic reasoning and reactive execution: longer reasoning chains may improve task decomposition, but delayed corrective actions can be unsafe in contact-rich manipulation, dynamic scenes, or settings with rapidly changing observations.
>
> **[Q2] Active cross-reference between VLA section and the uncertainty quantification open problem**
>
> We thank the reviewer for this suggestion. We agree that the original discussion of aleatoric and epistemic uncertainty was somewhat disconnected from the later discussion of VLA-based control and safety. To address this, we revised Section 3.1 to more explicitly connect the uncertainty concepts to real-world robot control. In particular, we added cross-references to the VLA discussion, where we discuss how uncertainty estimates can be used to define safety-relevant control boundaries, such as confidence thresholds for action execution, deferral or human intervention when the model is uncertain, filtering of unsafe actions, and caution under distribution shift or ambiguous observations. This highlights that aleatoric and epistemic uncertainty are not only abstract mathematical concepts, but can also influence when a VLA policy should act, slow down, request feedback, or avoid executing a risky action.
>
> **[Q3] Notations ambiguity**
>
> We thank the reviewer for raising this point of confusion. We agree with the reviewer that the notation in the manuscript was not standard and may cause confusion. We have updated the notation to match the standard notation suggested by the reviewer.

---

### Review · Reviewer_Gg5A · 2026-06-16

**Summary Of Contributions:**

This survey organizes robot learning along three axes: representation learning, VLAs, and world models. It then argues that five open challenges (uncertainty quantification, OOD generalization, cross-embodiment transfer, long-context understanding, long-horizon prediction) stem from the lack of integration across these components. The survey then goes into detail on the SOTA across these three access and ends with future unficiation.

**Audience:**

Yes

**Audience Explanation:**

I think the paper, once corrected for the requested changes, could serve as a good reference as a synthesis spanning representation learning, VLAs, and world models.

**Broader Impact Concerns:**

A short Broader Impact Statement noting the safety implications of real-world robot deployment would be a reasonable addition, but I do not consider it necessary for acceptance.

**Claims And Evidence:**

Yes

**Claims Explanation:**

Overall, the coverage is comprehensive, with a clear taxonomy and good supporting figures. Each section ends with a takeaway and a discussion part that a reader can refer to for deep dives. The world models section has a sharp critique, and Figure 8 backs it up well. Table 2 is a useful synthesis of the literature.

**Requested Changes:**

### Central Thesis and Layout
The paper’s central thesis appears to be that the five challenges are at least partly caused or exacerbated by the lack of integration. However, the paper does not precisely argue for the need to integrate them. After reading it, I was not clear on how the 5 challenges can be resolved through integration, or what failures could be specifically attributed to the lack of integration rather than to task difficulty. Section 7, where I expected this to be resolved, seems to me to be a catalog of patterns and future directions. Please provide a concrete map of failure modes and their integration fixes (e.g., a table) that states, for each challenge, what the failure mode is and how integration would resolve it.

Similarly, ‘integration’ is never defined concretely. It could refer to a wide range of technical choices, such as sharing representations, jointly training perception, policy, and prediction, using world models for VLA policy improvement, and so on. Please provide a taxonomy of integration types and use it consistently throughout the paper. This is also necessary to give context to the open problems later identified. On a related note, the paper also does not propose how to measure integration. Given the emphasis on benchmarks and real-world gaps, how would one evaluate a genuinely integrated system against a loosely coupled pipeline?

How did the authors decide what to include and what not to include? What was their search strategy, and what time window of literature did they consider? Please clarify these. Additionally, for Table 1, state the labeling rules more clearly: What qualifies as primary focus vs partial coverage? Finally, please justify why the selected axes are the correct axes for this categorization. This should be connected to the integration thesis that is central to the paper, and should justify why this is better than some other potential categorization (E.g. control, planning, Imitation Learning, RL, sim2Real, etc)

### Formalization
The current formalization in section 2 does not capture all that the paper later discusses. At times, it even contradicts the methods the paper surveys. I think the authors should rewrite section 2 backward from the 5 open problems, introducing exactly the objects required by each. Specifically:
- The notation $s$ conflates the true state, observation, and learned representation. Please distinguish the true state ($x_t$), observation $o_t$, representation $z_t$, history $h_t$, and belief $b_t$. Frame the problem as a partially observed decision process, thereby making the belief $b_t$ unify uncertainty quantification, long-context understanding, and object permanence. After the prelims, define the challenge formally in section 3 rather than narratively, thereby making these objects explicit.
- The one-step map $f(s,c) \to a$ does not capture closed-loop sequential, or generative policies. Please use a history conditioned policy as the object, such as $\pi_\theta (a_t \mid h_t,c)$, or $\pi_\theta (a_{t+H} \mid h_t,c)$ for action chunking.
- The MSE objective for imitation learning covers only deterministic behavior cloning and excludes the generative objectives the survey centers on in Section 5 (negative log-likelihood, diffusion denoising, flow matching, token/sequence prediction, inverse-dynamics, goal-conditioned, advantage-weighted imitation). Replace it with a general likelihood objective.
- The RL formulation does not define transition dynamics or distinguish the policy, environment, and trajectory distributions. Since Section 6 covers offline RL and world-model rollouts, the behavior-policy / learned-policy / dataset-distribution distinctions are needed for that material to be formally meaningful.
- The definition $g(s,c) \to s’$ collapses Language and action conditioning. These are different objects: a language-conditioned model predicts plausible futures satisfying an instruction, $p(s' | s, \text{instruction})$, whereas an action-conditioned dynamics model predicts the consequence of an intervention, $p(s' | s, a)$. This interventional distinction is the correlation-vs-causation point that the paper makes in Section 6

### Open Problems
Are the 5 problems identified exhaustive and independent? At the moment, they overlap. For example, the write-up uses statements such as ‘building on the challenge OOD generalization …' while treating these problems as parallel. As a result, cross-embodiment seems to be a special case of OOD; long context and long horizon are two sides (history/belief vs future/planning) of the same temporal problem. Please address this explicitly. Additionally, I recommend the following:
- **Cross-Embodiment:** distinguish what is being transferred (unified action spaces, object motion, skills, etc.)  using a source-target matrix.
- **History-conditioning:** The point about history-conditioned decision-making should more carefully distinguish between policies based on Markov states and those based on observations. The discussion should also separate memory, temporal abstraction, and action chunking. Action chunking improves temporal consistency in action generation, but it does not, by itself, solve long-context understanding or belief-state tracking.
- **Long-horizon prediction:** It currently discusses prediction, planning, task decomposition, and closed-loop execution together. These are different technical challenges. For example, the long-horizon reasoning for robotics may require predicting task-relevant variables instead of high-fidelity video prediction over long time spans
- Add a table mapping each challenge to its failure modes across the three components (perception, VLA, world models). This is distinct from Table 2, which maps works to challenges.

### Individual Domain Section  (Env, VLA, World Models)
At the moment, these sections read as standalone taxonomies rather than as support for the central pitch. I think they can be further refined by using the lens around integration to each one:
- **Env - Sec 4:** Evaluate each representation class by what it enables downstream, and not only what it captures. For example, does a given representation primarily improve observability, action selection, physical grounding, uncertainty estimation, temporal persistence, or world-model rollout quality? Figure 3 already lists some of these informally; systematize them into a functional comparison.
- **VLA- Sec 5:** Compare VLA families along action-centric dimensions (action representation, training objective, temporal structure, embodiment dependence, closed-loop feedback, physical grounding). As written, the section reads like a standard VLA survey, and its role in the overall survey is unclear.
- **World Models - Sec 6:** I think this is pretty strong, but it should be revised to focus on the question: How does the world model couple policy and representation? I recommend a table along the lines of: what the model predicts (pixels/latent/state/reward/task-relevant variables), how it couples with the policy, and what it enables downstream (data generation, planning/MPC, policy optimization in imagination, uncertainty estimation, safety). The correlation-vs-causation critique should be one column of this table, applied across the surveyed works.

### Reinforcement Learning
RL is a very central part of the world-model material, but in the current draft, it is treated lightly. Offline, online, and model-based RL do not receive as much grounding as the imitaiton and VLA method. While I think revising the formalization along the lines above partly addresses this, I also think some other lines of work in RL could be helpful. For example, the contextual-MDP framing can be used to formalize task or embodiment as context, which could also give the OOD and cross-embodiment challenges in Section 3 a precise basis. See Additional references for suggested papers.

Section 7 gestures at integrating structure ("object-centric reasoning, or physics-informed priors") as a future direction. I think this topic has also been studied in the RL literature, specifically looking at structured representations and inductive biases in deep RL, and could help support the arguments in section 7.

### Formatting
- Please use \citep and \citet; the inline citations are currently unreadable.
- Please run the paper through grammar-correction software and proofread it;  there are numerous typos and grammatical errors throughout.

### Additional References
- Ha & Schmidhuber (2018), World Models, arXiv:1803.10122. The eponymous world-model paper does not appear in the bibliography; for a survey centered on world models, this is a major omission. (Optionally, Schmidhuber's earlier 1990 work on predictive world models for priority.)
- Sutton & Barto (2018), Reinforcement Learning: An Introduction, 2nd ed., is a canonical reference for the distinctions between transition dynamics and distribution.
- Open problems as context inference
    - Hallak, Di Castro & Mannor (2015), Contextual Markov Decision Processes, arXiv:1502.02259:  formalizes task/embodiment as a latent context and is a special case of a POMDP, which ties directly to the belief-state reformulation
    - Zhang, Grefenstette & Rocktäschel (2021), A Survey of Generalisation in Deep Reinforcement Learning, arXiv:2111.09794: A unifying formalism for the OOD/cross-embodiment generalization discussion.
- Ni, Eysenbach, Seyedsalehi, Ma, Gehring, Mahajan & Bacon (2024), Bridging State and History Representations: Understanding Self-Predictive RL, ICLR 2024 (arXiv:2401.08898): shows that many state- and history-abstraction methods for MDPs and POMDPs are instances of a common self-predictive objective, directly supporting the requested distinction between state, observation, representation, and history, and the question of what the shared latent should encode (Section 7).

---

> ### Author Response · Authors · 2026-06-23
> **Author Response (1/5)**
>
> We thank the reviewer for the detailed and constructive feedback. We have carefully considered your feedback and  addressed each of your suggestions in the updated manuscript (The updated text is highlighted in blue in the revised manuscript). We provide brief responses to the feedback below:
>
> ### CENTRAL THESIS AND INTEGRATION:
>
> **[Q1]** We thank the reviewer for this important comment. We agree that the original version did not make the integration thesis sufficiently concrete. Our intended argument is not that the five challenges are caused only by lack of integration, or that integration alone resolves all task difficulty. Rather, our point is that several recurring failure modes become more severe when perception, action, and prediction are optimized as isolated components. We have revised the manuscript to make this distinction explicit.
> In the revised version, we now define integration more concretely and use this definition throughout the paper. In Section 2, we added a taxonomy of integration type. This provides a consistent vocabulary for discussing how different components can be coupled.
> We also revised Section 7 to directly address the reviewer’s concern by adding a new table (Table 6) that maps each of the five open challenges to its characteristic failure mode in loosely coupled systems, the components involved, and the integration mechanisms that can mitigate the failure.
> We also revised the surrounding Section 7 discussion so that the table is not merely a catalog of challenges, but a diagnostic map of how weak integration manifests as system-level failures. The revised text clarifies that integration should be evaluated not only by whether a system contains perception, policy, and prediction modules, but by whether coupling among these modules improves behavior under the relevant failure modes.
>
> **[Q2]** We thank the reviewer for this suggestion. We agree that the original manuscript used the term “integration” too broadly and did not provide enough concrete guidance on what kinds of technical coupling we meant. In Section 2 of revised manuscript, we now define integration as mechanisms that couple representation learning, VLA policies, and world models rather than optimizing them as isolated components. We also clarify that integration does not necessarily require a single monolithic architecture: a modular system can still be integrated if information, training signals, predictions, uncertainty estimates, or feedback from one component influence another in a task-relevant way.
> We added a taxonomy of integration types in Section 2, and use these terms consistently throughout the paper. Section 7 now refers back to these integration types when mapping each open challenge to its failure modes and possible integration mechanisms. We also use the same framing in the revised environment representation, VLA, and world-model sections to explain how each domain contributes to the broader integration argument. In addition to the discussion mapping different open challenges to their failure modes in Section 7, as suggested by the reviewer, we also add a discussion on how this integration may be evaluated. Rather than proposing a single scalar “integration score,” we frame integration evaluation in terms of whether coupling among components improves system-level behavior under the relevant failure modes.
>
> **[Q3]** We thank the reviewer for pointing this out. We agree that the original manuscript did not sufficiently explain how papers were selected, how Table 1 was labeled, or why the survey is organized around representation learning, VLA models, and world models. We have revised the introduction and table caption to make these choices explicit.
> First, we added a scope and search-strategy paragraph at the end of Section 1, clarifying that the survey is intended to be representative rather than exhaustive. We now specify the main venues and sources considered, the approximate time window of the literature, the keywords, and inclusion criteria used to identify relevant work.
> Second, we revised the caption and discussion around Table 1 to clarify the labeling rules. A method is marked as a primary focus when the corresponding axis is a central contribution or evaluation target of the work. It is marked as partial coverage when the axis is present or used as a supporting component, but is not the main contribution.
> Finally, we add a clearer justification for the choice of axes, and highlight that alternative categories such as control, planning, imitation learning, reinforcement learning, and sim-to-real are important, but they cut across these three axes rather than replacing them. We therefore believe the selected axes better support the paper’s central goal: understanding how perception, action, and prediction should be integrated in modern robot learning systems.

---

> ### Author Response · Authors · 2026-06-23
> **Author Response (2/5)**
>
> ### FORMALIZATION:
>
> We thank the reviewer for this detailed feedback. We agree that the original Section 2 was simplified and did not capture the history-conditioned, generative, and offline/model-based methods surveyed in Sections 5 and 6. We have substantially revised Section 2 to better align the preliminaries with the five open challenges and with the methods surveyed in later sections.
>
> We reformulate the robot learning setting as a partially observed decision process, allowing the same notation to support several later challenges and methods. We replaced the one-step observation-to-action map with a history-conditioned policy formulation to better match modern VLA policies, diffusion/flow/action-token policies, and closed-loop sequential decision-making systems discussed in Section 5. We replace the deterministic MSE-only imitation learning objective with a more general likelihood-based formulation over demonstrated actions or action sequences to better capture the range of imitation and generative policy objectives discussed in the survey. We revised the reinforcement-learning formalization to define the difference between online RL, offline RL and model-based RL to make  the formalization more compatible with the world-model material, while keeping the RL discussion concise so that it supports rather than dominates the survey. Finally, we revise the world-model formalization to distinguish language-conditioned future prediction from action-conditioned dynamics prediction. As pointed by the reviewer, this distinction is important for the correlation-versus-intervention issue discussed in Section 6: visually plausible future prediction is not necessarily the same as predicting the effect of a robot intervention. We believe these changes make the formalization more consistent with the rest of the paper and provide a stronger foundation for the open problems discussed later.
>
> ### OPEN PROBLEMS:
> We thank the reviewer for pointing this out. We agree that the five challenges should not be presented as exhaustive or mutually independent categories. In the revised manuscript, we now explicitly state that these challenges are representative open problems that often overlap and interact with one another, rather than a complete or orthogonal taxonomy. We revised the opening of the Section 3 to make these relationships clear. These revisions allow the open problems to function as a structured discussion of overlapping system-level challenges, rather than as an exhaustive list of independent problem categories.
>
> **[Q1] Cross-Embodiment**
>
> We thank the reviewer for this suggestion. We agree that the original cross-embodiment discussion did not clearly distinguish what is intended to transfer across embodiments. In the revised manuscript, we now clarify that cross-embodiment transfer is not simply the transfer of a source robot’s low-level actions to a target robot, but the transfer of  task objective, skill intent, or environment-level outcome, while the embodiment-specific means of execution must be adapted to the target robot’s morphology. To make this distinction explicit, we revise the opening of the cross-embodiment section, and frame cross-embodiment generalization as a structured form of OOD generalization. We considered the suggested source-target matrix, but chose to express this distinction in prose rather than as a separate table because the relevant transfer object varies across methods and is often not cleanly separable into a single source-target mapping. Some works attempt to align action spaces, some transfer object-centric effects, some use skill or motion abstractions, and others rely on embodiment-conditioned policies. A compact matrix would therefore risk suggesting sharper categories than the literature currently supports. Instead, the revised text emphasizes the main conceptual distinction needed for the survey: cross-embodiment transfer requires preserving task-level or environment-level intent while re-grounding execution in the target embodiment. This framing also connects directly to the paper’s integration thesis, since it highlights the need for representations and policies that separate embodiment-invariant task structure from embodiment-specific control.

---

> ### Author Response · Authors · 2026-06-23
> **Author Response (3/5)**
>
> **[Q2] History-Conditioning**
>
> We thank the reviewer for this clarification. We agree that the original discussion needed to more carefully distinguish Markov state information from partial observations and history-conditioned decision-making. We have revised both the preliminaries and the long-context understanding section to make this distinction explicit. In the revised formalization, we distinguish the true Markov state $x_t$ from the robot’s observation $o_t$, interaction history $h_t$, learned representation $z_t$, and belief $b_t$. We also revise the long-context discussion to avoid conflating history-conditioning with other temporal mechanisms. In particular, we now frame long-context understanding as the problem of retaining and using task-relevant past information through a history, memory, learned representation, or belief state. To keep the section focused, we now emphasize memory and belief maintenance rather than treating action chunking as a solution to long-context understanding. This also helps to connect  the challenge more directly to the integration thesis, since long-context understanding requires representations that preserve task-relevant history and policies that can use that history during closed-loop decision-making.
>
> **[Q3] Long-Horizon Prediction**
>
> We thank the reviewer for raising this point of confusion. We agree that long-horizon robot behavior involves several related technical mechanisms including prediction, planning, task decomposition, and closed-loop execution. We highlight that our goal of discussing these different works in Section 3 is not to provide a separate taxonomy of all of the challenges related to long-horizon prediction, but as examples to motivate long-horizon planning as a broad challenge that arises when robots must predict and reason over predicted actions and futures over long horizons under high-level instructions. We discuss specific mechanisms inline where they are most relevant throughout the VLA, World Model, and integration discussions, including hierarchical decomposition in VLA systems, future trajectory prediction and inverse-dynamics-based execution in world-model approaches, and policy-world coupling in Section 7. To avoid suggesting that these mechanisms are identical, we revised the phrasing to indicate that recent works address different aspects of the long-horizon challenge.
>
> ### INDIVIDUAL DOMAIN SECTIONS
>
> **[Q1] Env. Representation**
>
> We thank the reviewer for this helpful suggestion. We agree that Section 4 should not only describe what different environment representations capture, but also analyze what they enable downstream for robot learning. In the revised manuscript, we added a functional comparison table (Table 4) that evaluates representation classes according to their downstream roles.
> Specifically, Table 4 compares the different representations along the dimensions suggested by the reviewer. We also revised the surrounding discussion to emphasize that representations act as interfaces between perception, policy learning, and predictive modeling. We clarify in the table caption that the labels are not strict guarantees, but indicate typical capabilities supported by the representation’s native structure and by how such representations are commonly used in the surveyed literature. This makes Section 4 more directly connected to the paper’s integration thesis: representation learning is evaluated not only by perceptual fidelity, but by whether the resulting representation supports downstream action selection, uncertainty estimation, temporal reasoning, and world-model prediction.

---

> ### Author Response · Authors · 2026-06-23
> **Author Response (4/5)**
>
> **[Q2] VLA**
>
> We thank the reviewer for this helpful suggestion. We agree that the VLA section should clearly support the paper’s broader integration argument rather than read as an isolated taxonomy of recent VLA models. The role of Section 5 in our survey is to examine the policy/action-grounding axis of modern robot learning. While Section 4 discusses what environment representations encode and enable, Section 5 discusses how vision-language representations are transformed into executable robot actions, and Section 6 discusses how predictive/world models estimate future consequences. Thus, the VLA section is included not simply as a standard survey of VLA architectures, but to analyze how different model families connect perception, language, and action.
>
> The action-centric dimensions highlighted by the reviewer are already discussed throughout the section, but we agree that their connection to this role could be made clearer. Specifically, we discuss action representation in the treatment of action tokens, continuous actions, action chunks, normalized action spaces, object motion, and pixel motion; training objectives in the discussion of supervised action prediction, imitation learning, diffusion policies, flow matching, auxiliary alignment losses, and safety-aware imitation; temporal structure in the discussion of action chunks, reasoning traces, long-horizon task decomposition, past key-frame conditioning, and spatiotemporal VLA representations; embodiment dependence in the discussion of cross-robot datasets, normalized action spaces, hardware-specific prompts, and motion-centric policies; closed-loop feedback in the discussion of dynamic VLAs, iterative refinement, asynchronous execution, uncertainty-triggered intervention, and reward feedback from world models; and physical grounding in the discussion of 3D-VLA methods, depth-aware VLA models, point-cloud-conditioned action heads, object-motion prediction, pixel-motion representations, and safety constraints.
>
> To make this action-grounding role clearer in the revised manuscript, we revise the opening of Section 5 to frame VLA models as policies that transform interaction histories, visual representations, and language/task context into executable robot actions. We also added clarifying text throughout the section to emphasize how different VLA families ground vision-language representations into actions. Finally, we revise the discussion to explicitly connect these design choices to integration, emphasizing how VLA systems couple action representations, temporal structure, embodiment constraints, uncertainty estimates, feedback, and physical grounding.
> We chose not to add a separate exhaustive comparison table for these dimensions because the section already discusses them inline where they are most relevant. Instead, the revised prose makes the action-grounding role of VLA models explicit while preserving the section’s organization around the major architectural paradigms used in the literature.
>
> **[Q3] World Models**
>
> We thank the reviewer for this suggestion. We agree that Section 6 should more explicitly emphasize how world models couple representations to policies, rather than only organizing works by model type. In the revised manuscript, we reframe the opening and discussion of Section 6 around this role.
> We also add a new representative table (Table 5) in Section 6 following the structure suggested by the reviewer. The table summarizes representative world models according to what they predict, how those predictions are coupled to policies, what downstream use they enable, and what coupling limitation remains. We also incorporated the reviewer’s correlation-versus-causation critique into the limitation column of the table and the surrounding discussion.
> Finally, we revised the Section 6 discussion to clarify that world models are increasingly used not only for prediction, but also as components of policy optimization, planning, safety filtering, evaluation, and data generation. This makes their reliability especially important: if predicted futures do not preserve object permanence, physical constraints, or the effects of specific interventions, policies may optimize against an inaccurate imagination space. We believe these changes make Section 6 more directly support the paper’s central integration thesis.

---

> ### Author Response · Authors · 2026-06-23
> **Author Response (5/5)**
>
> ### REINFORCEMENT LEARNING
>
> We thank the reviewer for highlighting this point of concern. We agree that the original manuscript treated the reinforcement-learning foundations of world models too lightly, especially given the role of offline, online, and model-based RL in the world-model literature. In the revised manuscript, we strengthen the RL grounding in several places, including the background formulation in Section 3, while keeping the discussion proportional to the survey’s broader focus on representation learning, VLA policies, and world models.
>
> We incorporate the reviewer’s point about structured representations and inductive biases in the future-directions discussion. We now note that object-centric abstractions, relational representations, factorized dynamics, and physics-informed models have also been studied in reinforcement learning as ways to improve exploration, value estimation, transfer, and model-based planning. From the integration perspective of the survey, these ideas are important because representations should not only encode perceptual similarity, but also expose entities, relations, constraints, and action effects that are useful for both policy learning and world-model prediction.
> Overall, these revisions strengthen the RL grounding where it is needed for the paper’s argument, while keeping the manuscript focused on its central integration thesis rather than expanding it into a separate survey of reinforcement learning.
>
> ### FORMATTING
>
> We thank the reviewer for raising this point of concern regarding the readability of the citations, as well as regarding the typographical errors in the manuscript. We have revised the manuscript to appropriately use \citep and \citet to improve the readability, and  to fix any typos and grammatical errors.
>
> ### ADDITIONAL REFERENCES
>
> We thank the reviews for directing us to these relevant references.  We have incorporated some of the suggested references including the ones on world models, RL and contextual MDPs in relevant parts of the manuscript.